# Smoking Habit and Respiratory Function Predict Patients’ Outcome after Surgery for Lung Cancer, Irrespective of Histotype and Disease Stage

**DOI:** 10.3390/jcm12041561

**Published:** 2023-02-16

**Authors:** Davide Piloni, Francesco R. Bertuccio, Cristiano Primiceri, Pietro Rinaldi, Vittorio Chino, David Michael Abbott, Federico Sottotetti, Chandra Bortolotto, Francesco Agustoni, Jessica Saddi, Giulia M. Stella

**Affiliations:** 1Cardiothoracic and Vasculat Department, Unit of Respiratory Diseases, IRCCS Policlinico San Matteo, 27100 Pavia, Italy; 2Medical School, University of Pavia, 27100 Pavia, Italy; 3Cardiothoracic and Vascular Department, Unit of Thoracic Surgery, IRCCS Policlinico San Matteo, 27100 Pavia, Italy; 4Department of Surgical, Pediatric and Diagnostic Sciences, University of Pavia, 19001 Pavia, Italy; 5Department of Medical Oncology, IRCCS ICS Maugeri, 27100 Pavia, Italy; 6Department of Diagnostic Services and Imaging, Unit of Radiology, Fondazione IRCCS Policlinico San Matteo, 27100 Pavia, Italy; 7Department of Internal Medicine and Medical Therapeutics, University of Pavia Medical School, 27100 Pavia, Italy; 8Department of Oncology, Unit of Oncology, Fondazione IRCCS Policlinico San Matteo, 27100 Pavia, Italy; 9Medical School, Milano-Bicocca of University, 20900 Monza, Italy; 10Department of Oncology, Unit of Radiation Therapy, Fondazione IRCCS Policlinico San Matteo, 27100 Pavia, Italy

**Keywords:** NSCLC, thoracic surgery, predictors, data mining, personalized medicine

## Abstract

Background. Growing evidence suggests that sublobar resections offer more favorable outcomes than lobectomy in early-stage lung cancer surgery. However, a percentage of cases that cannot be ignored develops disease recurrence irrespective of the surgery performed with curative intent. The goal of this work is thus to compare different surgical approaches, namely, lobectomy and segmentectomy (typical and atypical) to derive prognostic and predictive markers. Patients and Methods. Here we analyzed a cohort of 153 NSCLC patients in clinical stage TNM I who underwent pulmonary resection surgery with a mediastinal hilar lymphadenectomy from January 2017 to December 2021, with an average follow-up of 25.5 months. Partition analysis was also applied to the dataset to detect outcome predictors. Results. The results of this work showed similar OS between lobectomy and typical and atypical segmentectomy for patients with stage I NSCLC. In contrast, lobectomy was associated with a significant improvement in DFS compared with typical segmentectomy in stage IA, while in stage IB and overall, the two treatments were similar. Atypical segmentectomy showed the worst performance, especially in 3-year DFS. Quite unexpectedly, outcome predictor ranking analysis suggests a prominent role of smoking habits and respiratory function, irrespective of the tumor histotype and the patient’s gender. Conclusions. Although the limited follow-up interval cannot allow conclusive remarks about prognosis, the results of this study suggest that both lung volumes and the degree of emphysema-related parenchymal damage are the strongest predictors of poor survival in lung cancer patients. Overall, these data point out that greater attention should be addressed to the therapeutic intervention for co-existing respiratory diseases to obtain optimal control of early lung cancer.

## 1. Introduction

In recent years more lung cancer diagnoses have occurred owing to the improvement of radiological techniques and the introduction of low-dose computed tomography in screening campaigns [1]. The optimal therapeutic approach for early-stage NSCLC (I–II, and, in some specific cases, IIIa) is lung resection surgery with mediastinal lymphadenectomy. In patients with an adequate respiratory reserve, lobectomy has been the gold standard, whereas segmentectomy was reserved for unfit patients [2,3]; however, recent results of the CALGB and JCOG/WJOC noninferiority trials clearly showed the oncological efficacy of a sublobar resection in early-stage disease [4,5,6,7]. The significant reduction in residual capacity after the removal of a lung lobe in patients with COPD, with a resulting increase in perioperative mortality and a reduced life expectancy, has been reported in the literature for years and represents a patient population that could benefit from sublobar resection [8,9]. The CALGB and JCOG/WJOC trials could change current practice, but they leave some problems unsolved: (i) not all the segments are equally easy to resect, and thus technical issues may arise; (ii) costs may be higher if adjuvant treatments/additional surgeries are needed (indeed, local recurrence in the segmentectomy group was nearly double that of the lobectomy group); (iii) lung function preservation may also not be significant enough to justify the risks [4,10]. Some papers have, however, reported that the incidence of local and distant cancer recurrence seems not to be influenced by the extension of parenchymal resection [11], but a percentage of patients, varying from 30% to 55%, develops disease recurrence and eventually dies despite an initial resection with curative intent [12,13]. Thus, a more efficient identification and selection of those lung cancer patients who will really benefit from surgery is needed. In this paper we aim to compare the two different surgical approaches through a five-year retrospective study to identify predictive and prognostic markers.

## 2. Materials and Methods

### 2.1. Patient Identification and Selection

The study retrospectively analyzes data from 153 patients with NSCLC in clinical stage 8th edition TNM I who underwent a pulmonary resection with hylo-mediastinal lymphadenectomy at the Thoracic Surgery Unit of IRCCS San Matteo Hospital. The time interval of the study is between January 2017 and December 2021, with an average follow-up time of 25.5 months. Exhaustive clinical data regarding the population studied as well as the histological subtypes and disease stage distribution are reported in Table 1. Informed consent of each patient was collected routinely at hospital admission in accordance with standard hospital procedures. Patients’ data were collected through consultation of operating directories; oncological, pneumological, and pathological reports; and discharge letters. Patients were divided into three groups based on the type of resection performed: (i) lobectomy, (ii) typical segmentectomy, and (iii) atypical segmentectomy. Inclusion criteria for each surgical approach were defined based on (i) respiratory function, (ii) comorbidities, and (iii) surgeon’s judgment. Of the patients, 103 underwent lobectomy, while 26 and 24 patients underwent typical and atypical segmentectomy, respectively. Patients with idiopathic pulmonary fibrosis and other interstitial lung diseases with severe impairment of lung function were excluded from surgery, as were subjects affected by other severe comorbidities, after a multidisciplinary evaluation.

#### Preoperative Studies

All 153 patients evaluated in the study underwent preoperative studies, including blood tests and imaging examinations (total-body contrast medium CT and PET), to assess the disease extension. Regarding respiratory function tests, the predicted postoperative FEV1 and DLCO were calculated according Kearney’s equations [14]. In cases when predicted values suggested a risk of respiratory complications after surgery, the patients were sent to obtain ventilation/perfusion scintigraphy as a second-level study [15,16,17]. Maximum oxygen consumption (VO2 max) via cardiopulmonary exercise testing (CPET) [18,19,20] is not available in our institution (patients requiring this type of investigation were sent to other institutions and excluded from the present study). Cito-histologic diagnosis was obtained through transthoracic CT-guided needle aspiration, EBUS-TBNA, and endobronchial biopsy. All cases were discussed within the Interdisciplinary Group of Thoracic Neoplasms (GINT) at IRCCS San Matteo Hospital Foundation for proper diagnosis consensus, staging, and clinical management.

### 2.2. Surgical Techniques

The open approach was performed in 40 patients (26.14%), while VATS was used in the remaining 113 patients (73.86%). In the group of patients subjected to the open approach, a clear correlation with lobar resections, about 38 out of 40 total (95%), was documented. The typical and atypical segmentectomy procedures were performed as already defined [21,22]. The distribution of resected segments and lobes is provided in Appendix A. For all the techniques, systematic ilo-mediastinal lymph node dissection was performed at the end of the lung resection. Management of chest drains and discharge planning was individualized based on the patient’s clinical characteristics and the surgeon’s judgment. In case of prolonged air leaks, patients were discharged after placing a Heimlich valve and ensuring the stability of lung expansion. Discharge criteria were not influenced by the type of surgery performed.

#### Surgical, Postsurgical, and Follow-Up Data

For all three groups of patients, the data derived from the operating registers and pathology reports were recorded. This data included histotype, infiltration of the surgical margin, the distance between the neoplasm and the surgical resection margin, the lobe or segment affected by the resection, and the number and type of lymph nodes resected. R0 resection (disease-free surgical margins) was defined by pathologists as the macroscopic and microscopic absence of neoplastic cells on the resection margins. Based on the distance in centimeters between the tumor and the resection margin, three categories of patients were created: distance >2 cm, distance between 1 and 2 cm, and distance <1 cm. The follow-up was performed at the Respiratory Diseases or Oncology units and consisted of clinical examination every three months, respiratory function tests, and execution of total-body CT scan with contrast medium. If these exams provided suspicion of disease relapse, subsequent diagnostic procedures were defined by an interdisciplinary discussion. Survival analysis was carried out by taking into consideration two end points: OS and DFS. OS was calculated as the time elapsed between the surgery and the date of death or of the last follow-up. The DFS was calculated as the time in months that elapsed between the surgery and the development of relapse.

### 2.3. Statistical Analysis

Statistical analysis was carried out using SPSS 22 (IBM SPSS statistics 20 IBM Corporation, Chicago, IL, USA) and JASP software. The continuous variables were expressed by means of the mean value ± the standard deviation (SD), and the latter was compared using the Student’s *t* test for independent variables; the nominal variables were compared by means of χ^2^ tests. The Kaplan–Meier method was used to create the cumulative survival curves, while the log-rank test was used to calculate the differences between the curves. A *p* value < 0.05 was considered statistically significant. The entire dataset was then analyzed using the JMP partition algorithm (JMP-Statistical Discoveries. From SAS, website at www.jmp.com), which can search for all possible subdivisions of the best predictors of response/outcome. These data splits (or partitions) were performed recursively to form a tree of decision rules as already described [23]. It should be noted that to determine how much a predictive model is really able to describe reality, the validation methods differ according to whether they use regression or estimate a number or classification that associates each element with the categories to which it belongs. While for the regression we need to evaluate the difference between two numbers (the observed value and the predicted one), in the classification approach it is sufficient to count the number of times in which the class attributed to the model (our target) is the right one, namely, the one observed. Because the cohort evaluated is small, the regression model cannot be validated.

## 3. Results

The mean age of patients undergoing lobectomy was 68.5 (±9.02), while 70.8 (±6.71) and 71.68 (±8.98) were, respectively, the ages of patients undergoing typical segmentectomy and atypical segmentectomy. Patients with stage IA had a mean age of 69.17 (±8.9) and those with IB were 70.1 (±8.15) years old on average. Regarding the numbers and percentages of relapses based on the tumor stage, considering stage IA globally, 20 relapses out of 118 patients (17%) occurred. Specifically, 10 out of 77 (13%) and 4 out of 20 (20%) relapses occurred in patients undergoing lobectomy and segmentectomy, respectively, while 6 out of 21 (28.6%) relapses occurred in patients undergoing AS (*p* = 0.523). Overall, the highest recurrence rate was documented in stage IB, in which 8 out of 35 patients were positive (22.8%); specifically, 6 out of 26 in the L group (23%), 1 out of 6 for the TS group (16.6%), and 1 out of 3 in the AS group (33.3%) (*p* = 0.116). Details are available in Table 1, panel A. The most common resections were for the right upper lobectomy group (28 of 103 (27.18%)). Segmental resections of the left upper lobe (in 13 of 26 cases (50%)) were the second most common resections, and atypical segmental resections of the right upper lobe (in 11 of 24 cases (45.8%)) were the least common resections (Appendix A). The total number of resected lymph nodes was 8.19 ± 3.016 in patients undergoing lobectomy, 5.16 ± 2.55 in patients undergoing typical segmentectomy, and 3.91 ± 2.37 in patients undergoing atypical segmentectomy (*p* < 0.001). The total number of resected lymph node stations was 2.7 ± 1.0, 2.6 ± 0.94, and 1.75 ± 0.94 in patients undergoing lobectomy, typical segmentectomy, and atypical segmentectomy, respectively (*p* < 0.001) (Appendix A). The invasion of the surgical margin by the neoplasm (resection R1) was found in 6 cases (3 typical (11.5%) and 3 atypical (12.5%) segmentectomies) and in no case of lobectomy (0%) (Appendix A). A distance between the surgical margin and the resection margin <1 cm emerged in 22 of 103 cases of lobectomies (21.3%), 19 of 26 cases of typical segmentectomy (73%), and 14 of 24 cases of atypical segmentectomy (58 %). The presence of a distance between the surgical margin and the resection margin between 1 and 2 cm was present in 18 of 103 cases of lobectomies (17.5%), 3 of 26 cases of typical segmentectomies (11.5%), and 7 of 24 cases of atypical segmentectomies (29.2%). A distance >2 cm was found in 63 out of 103 lobectomy cases (61.16%) and 1 out of 26 and 0 out of 24 cases following typical and atypical segmentectomy (4% and 0%), respectively.

In the three cohorts examined, the total number of relapses was 28 out of 153 patients (18.30%) (*p* value = 0.2), with 16 out of 103 cases in the lobectomy group (15, 53%), 5 out of 26 patients in the typical segmentectomy group (19.23%), and 7 out of 24 in the atypical group (29.16%) (*p* value = 0.411) (Appendix A). In the lobectomy group, the frequency of relapse within 1 year was very low, about 18.75% (3 cases out of 16), compared with 60% in the typical segmentectomy group and 57.15% in the atypical group (*p* = 0.048). However, relapses between 4 and 5 years was higher in the lobectomy group at 25% versus 0% of the patients undergoing segmentectomy (*p* = 0.052). It is important to evaluate the frequency of recurrence also, according to the clinical stage of the patient. In this perspective, considering stage IA globally, 20 relapses out of 118 patients (17%) occurred; specifically, 10 out of 77 (13%), 4 out of 20 (20%), and 6 out of 21 (28.6%) in patients undergoing lobectomy, typical segmentectomy, and atypical segmentectomy, respectively (*p* = 0.523) (Table 1). Overall, the highest recurrence rate was documented in stage IB with 8 out of 35 patients being positive (22.8%); specifically, 6 out of 26 in the lobectomy group (23%), 1 out of 6 for the typical segmentectomy group (16.6%), and 1 out of 3 in the atypical segmentectomy group (33.3%) (*p* = 0.116). By using the Kaplan–Meier method, it was possible to estimate the disease-free survival (DFS) of all patients recruited into the study regardless of the surgery performed and the clinical stage (Table 1 and Appendix A); the DFS was 91% at 1 year, 77% at 3 years, and 57% at 5 years. Overall DFS was then estimated based on stage, disregarding the surgical treatment: for stage IA it was 93% at 1 year, 79% at 3 years, and 54% at 5 years; for stage IB it was 90% at 1 year, 69% at 3 years, and 69% at 5 years (*p* = 0.741). Moreover, Kaplan–Meier curves were created to evaluate the differences among the three different techniques. Taking both stages together, our data show that the DFS at 1 and 3 years in the patients undergoing lobectomy was slightly greater than, but comparable to, that of the patients undergoing TS (*p* = 0.125), but it was much greater than in the patients undergoing AS (*p* = 0.016). Regarding the two segmentectomy techniques, minor differences were observed, although TS has been shown to have a higher DFS at 1 and 3 years. The OS of the stage IA patients undergoing lobectomy was 100% at 1 year and 89.4% at 3 and 5 years, in the TS group it was 100% at 1 year, and in the AS group it was 100% at 1 year and 90% at 3 years. In the stage IB patients, on the other hand, OS in the lobectomy group was 95% at 1 year and 72.4% at 3 and 5 years; for the TS group it was 100% at 1 year and 80 % at 3 years; and finally, in the AS group, it was 100% at 1 year and 50% at 3 years (*p* = 0.266).

We then moved to evaluate the data through an unbiased approach focused on partition analysis and the ranking of predictors (Figure 1). In the patients undergoing surgery, both lobectomy and segmentectomy, the factor most associated with the outcome, defined as death at least 6 months after surgery, was the histotype. All patients who were affected by squamous cell carcinoma were alive (at the end of 6 months?). In the patients with adenocarcinoma, the second factor associated with survival was smoking: nonsmokers were more likely to die. Finally, male sex was associated with a worse outcome. Moreover, the partition analysis showed that smoking habit and respiratory functional tests significantly correlated with the patients’ outcome. Among smokers, male patients carrying stage IB tumors represented the subgroup with the worst prognosis. On the other hand, outcome among never smokers was affected by the patients’ respiratory functions. Finally, the ranking of predictor markers showed that pulmonary function and smoking habits significantly impacted the patients’ survival. Within the limits of the cohort analyzed, all the patients with poor performance on the respiratory function tests who were still alive were treated with inhaled bronchodilators. However, the relatively small size of the population did not allow comparison of the efficacy of dual and triple COPD therapy in terms of statistical significance.

## 4. Discussion

Many articles have compared the clinical performances of lobar vs. sublobar surgery in early-stage lung cancer, and recent data suggest that the second approach is superior in small-sized lung cancer (<2.0 cm in diameter with a relevant ground glass component) [4,5,24,25,26]. Although these results have created great interest among oncologists and surgeons and are going to change clinical practice, some details still need to be clarified. For instance, quite unexpectedly, the improved survival rates associated with segmentectomy seem to be related to a decrease in deaths related to cancer recurrence or second primary tumor arousal. Thus, a better identification/validation of the predictive and prognostic markers is needed. The results of the present study demonstrated that if stages IA and IB are considered together, the DFS at 1 and 3 years of patients undergoing lobectomy is slightly greater than, but comparable to, that of patients undergoing typical segmentectomy (*p* = 0.125) and is much greater than that of patients undergoing atypical segmentectomy (*p* = 0.016). Regarding the two segmentectomy techniques, minor differences were observed although typical segmentectomy has been shown to have a higher DFS at 1 and 3 years. When we controlled for treatment in stage IA patients, the OS was superior to that of stage IB patients (87.5% at 5 years vs. 70.5% at 5 years) (*p* = 0.023). Atypical segmentectomy was the treatment with the worst overall survival at 3 years. These data indicate that lobectomy involves a greater probability of obtaining a radical resection (R0), whereas both types of segmentectomy are associated with a greater probability of obtaining an R1 resection, though the difference was not statistically significant (*p* = 0.272) (*p* = 0.332). Moreover, lobectomy was associated with a statistically significant increase in the number of lymph nodes removed compared with both typical and atypical segmentectomy (*p* < 0.001). There is a small (although statistically significant) difference between lobectomy and atypical segmentectomy, mainly in terms of number rather than the lymph node stations removed. Overall, the extent of the lymph node dissection cannot be accounted for on the size of the resection, and no conclusions should be allowed regarding the association with disease relapse. In summary, this first set of results is coherent and reproduces the most recent published data regarding the rationale for sublobar surgery in small lung cancers.

The major finding of this work is otherwise related to the observation that smoking history and respiratory function can predict patients’ outcomes. These results point out that greater attention should be addressed to the therapeutic intervention for co-existing respiratory obstructive syndromes to obtain optimal control of early lung cancer. Partition analysis applied to our dataset pointed out some unexpected results. Although a detailed analysis goes beyond the scope of this work and will be further analyzed, some issues deserve a discussion. Firstly, we noticed that the adenocarcinoma subtype is associated with worse outcomes. This finding is only partly unexpected, as the literature reports that adenocarcinomas arising in nonsmoking patients present a multifocal growth pattern and are thus more associated with relapse after surgery [27,28,29]. Concurrently, it is highlighted that smoking habit in the analyzed population was not associated with higher mortality and that respiratory function significantly impacts patients’ outcome, thus emerging as a critical variable at each node in different decision trees. Interestingly, in both males and females the outcome is favorable in cases of reduced respiratory function. Despite the limited size of the cohort analyzed, these points clearly suggest that both the lung volume and the degree of parenchymal damage due to emphysema are predictive markers of decreased survival in lung cancer patients, irrespective of their gender, disease stage, and histotypes. Moreover, these findings point out two clinically relevant considerations: the role of early cancer diagnosis and the role of inhaled bronchodilators. First, smokers seem to be better screened, diagnosed, and treated, and this approach ultimately led to a better outcome when cancer was found. Second, patients with worse respiratory function are probably those who are treated or properly treated with bronchodilators. Within the limits of the cohort analyzed, all the NSCLC patients with poor respiratory test performance who were still alive were treated with inhaled bronchodilators even though we did not have a statistically significant comparison of the efficacy between dual and triple anti-COPD therapy in this specific population. Although this point goes beyond the scope of the study, its preliminary findings sustain a rationale for a better analysis of the role of bronchodilation in early-stage cancer patients. 

There are several limitations in this study, including the following: (i) it is a retrospective study with a high probability of selection and reporting biases; (ii) there was a short follow-up period for those that were operated on at the end of the year 2021; (iii) the cohort of patients undergoing typical and atypical segmentectomy is numerically scarce compared with the lobectomy cohort population. Notably, the OS of patients undergoing atypical segmentectomy may have been influenced by the high risk of death from comorbidities and the numerous risk factors that this cohort possesses. However, it should be underlined that patients with squamous cell carcinoma, even in early disease stage [30,31], generally have a poor prognosis because they have a history of heavy smoking and may have multiple other diseases.

## 5. Conclusions

Within the limits of the population analyzed, the results of the present study suggest that patients with larger tumors and general health conditions that can sustain larger resections should be treated with a lobectomy, as it allows for a wider margin of resection, thus creating more distance from the tumor. Typical segmentectomy is still a viable alternative strategy for stage I NSCLC with the intention of preserving better lung function. Because the atypical segmentectomy has proved to be slightly inferior to the typical one, it should be used in compromised patients only when a typical resection cannot be performed. These results suggest that both the lung volume and the degree of parenchymal damage from emphysema are predictive markers of death in lung cancer patients irrespective of their gender, disease stage, or cancer histotypes. Overall, these data point out that greater attention should be given to the therapeutic intervention in patients with coexisting obstructive respiratory syndromes to obtain optimal control of early lung cancer (Figure 2).

## Figures and Tables

**Figure 1 jcm-12-01561-f001:**
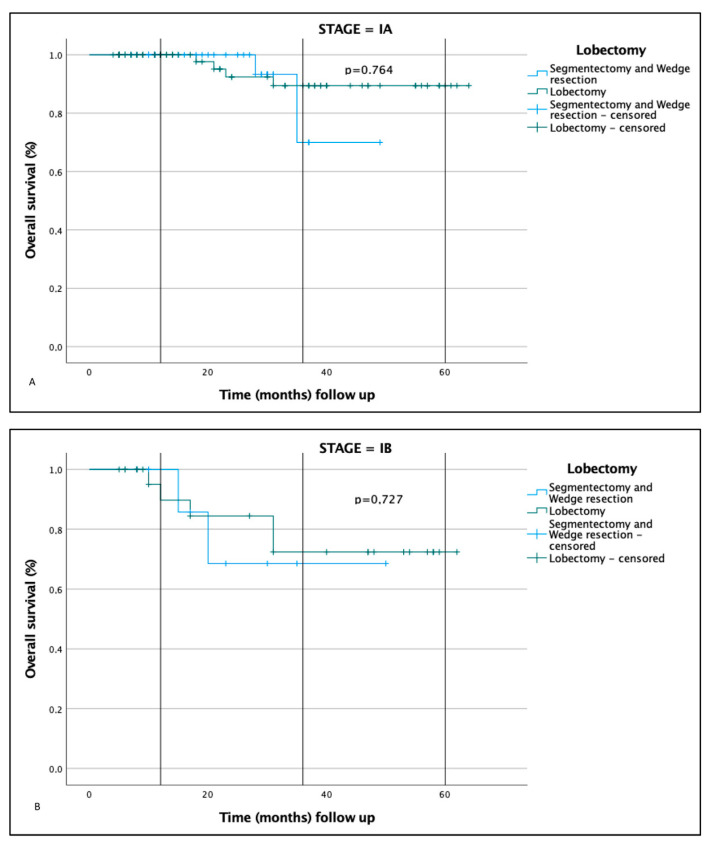
Overall survival (OS) (%) of all the enrolled patients. OS comparison between lobectomy and segmentectomy (typical and atypical) for disease stages IA (**A**) and IB (**B**), respectively.

**Figure 2 jcm-12-01561-f002:**
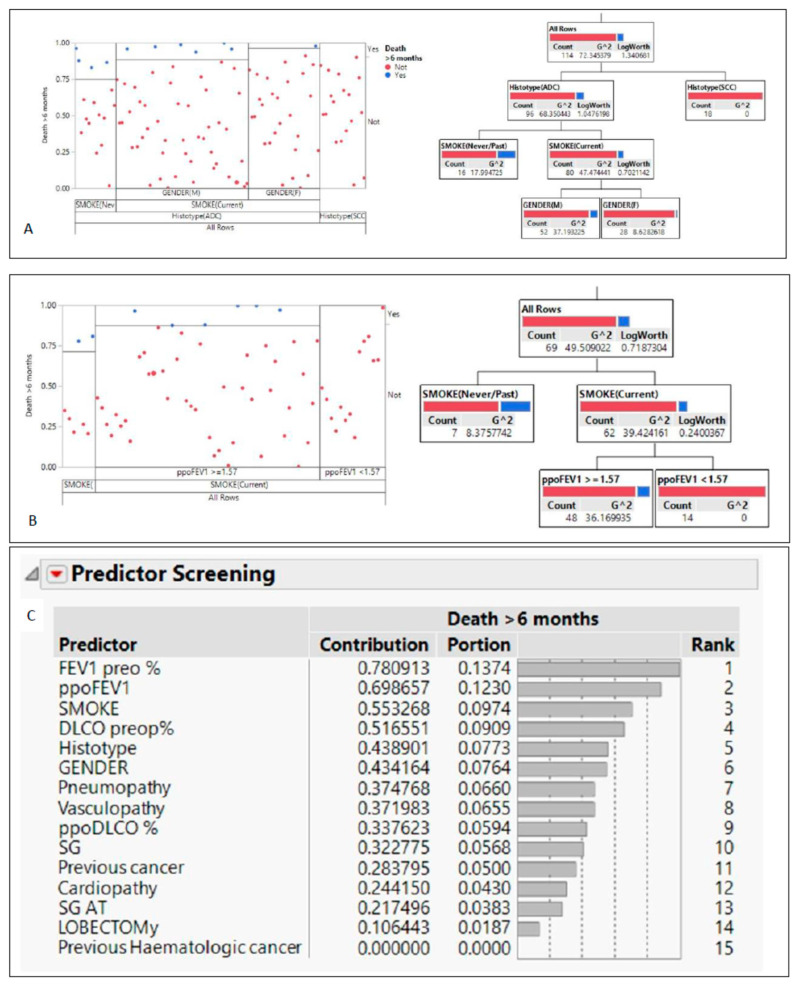
Partition analysis and variable ranking applied to the dataset. (**A**) In raw analysis of the whole cohorts undergoing surgery, squamous cell cancer and stage IA were the factors most associated with better outcomes; (**B**) smokers with the worst lung function and harboring adenocarcinoma had better survival than never smokers; (**C**) predictor ranking analysis applied to the whole cohort indicates FEV1 and smoking as the most relevant variables that impact the 6-month outcome.

**Table 1 jcm-12-01561-t001:** (**A**) Demographics and clinical and pathologic details of the three cohorts analyzed. (**B**) Disease-free survival (DFS) and overall survival (OS) data defined for the 1st, 3rd, and 5th year of follow-up for all the enrolled patients according to the Kaplan–Meier method.

(A)						
	Lobectomy	%	Typical SG	%	Atypical SG	%
Mean Age at Diagnosis (yrs)	68.5 (9.02)		70.8 (6.71)		71.58 (8.98)	
Males	60	58.25%	14	53.85%	17	70.83%
Females	43	41.75%	12	46.15%	7	29.17%
	103	100.00%	26	100.00%	24	100.00%
Smokers	85	82.52%	22	84.62%	23	95.83%
Never smokers	18	17.48%	4	15.38%	1	4.17%
Comorbidities						
Pneumopathy	40	38.83%	19	73.08%	12	50.00%
Cardiopathy	38	36.89%	15	57.69%	12	50.00%
Vasculopathy	35	33.98%	10	38.46%	11	45.83%
Previous neoplastic disease	27	26.21%	9	34.62%	10	41.67%
Previous hematologic cancer	9	8.74%	3	11.54%	2	8.33%
Others	51	49.51%	9	34.62%	10	41.67%
Disease stage						
IA	77	65.3%	20	17%	21	17.7%
IA1	26	18.45%	6	7.69%	3	16.67%
IA2	38	36.89%	12	46.15%	10	41.67%
IA3	20	19.42%	6	23.08%	7	29.17%
IB	26	25.24%	6	23.08%	3	12.50%
	103	100.00%	26	100.00%	24	100.00%
Histotype						
Adenocarcinoma	88	338.46%	21	350.00%	18	600.00%
Squamous cell carcinoma	15	57.69%	5	83.33%	6	200.00%
	103	396.15%	26	433.33%	24	800.00%
Relapses						
IA	10/77	13	4/20	20	6/21	28.6
IB	6/26	23	1/6	16.6	1/3	33.3
	16/103	15.5%	5/26	19.2%	7/24	29.2%
**(B)**						
**Year**	**1**	**2**	**3**	**1**	**2**	**3**
	**Disease-Free Survival (DFS) %**			**Overall Survival (OS) %**		
Whole data	**91**	**77**	**57**	**97**	**83**	**83**
Stage						
IA	93	79	54	100	87.6	87.6
IB	90	69	69	96.4	70.5	70.5
Surgery						
Lobectomy	97.5	81-4	62.2	97	84.5	84.5
Segmentectomy	82	77.7	60	100	94	94

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
