# Peer review of "Smoking Habit and Respiratory Function Predict Patients’ Outcome after Surgery for Lung Cancer, Irrespective of Histotype and Disease Stage"

_jcm, 2023, doi:10.3390/jcm12041561_

Round 1

Reviewer 1 Report (Previous Reviewer 2)

Thank you for your fine revision.

Author Response

We thank the Reviewer for the comments and for suggestions.

Reviewer 2 Report (New Reviewer)

This is a very well-written study. Despite its small sample size, the results are interesting. I have a few suggestions:

1) Firstly, given the title suggests that the major theme of the paper is the finding of smoking history and respiratory function predicting patient outcomes, the Discussion should be crafted to highlight and discuss this main result. Currently, its mention and reference seems to be in passing, right at the end of the Discussion.

2) It is customary to have the limitations paragraph at the end of the Discussion. Currently it is placed as the penultimate paragraph. Please shift it to the end.

3) While the partition tree analysis identifies smoking and respiratory function as top predictors, I wonder whether the authors considered using a traditional logistic regression model to see if those two factors remained significant predictors after adjusting for all possible confounders.

Author Response

Response to Reviewer 2

This is a very well-written study. Despite its small sample size, the results are interesting. I have a few suggestions

We thank the Reviewer for careful reading of the manuscript and for fruitful suggestions and comments.  Below point-by-point answers (A) to each comment (C)

C1) Firstly, given the title suggests that the major theme of the paper is the finding of smoking history and respiratory function predicting patient outcomes, the Discussion should be crafted to highlight and discuss this main result. Currently, its mention and reference seems to be in passing, right at the end of the Discussion.

C1) we thank the Reviewer for pointing out this criticism. The Discussion section has been revised as follows: “In summary this first set of results is coherent and reproduces most recent published data regarding the rationale of sublobar surgery in small lung cancers. The major finding of this work is otherwise related to the observation that smoking history and respiratory function can predict patients’ outcomes. These results point out that greater attention should be addressed to the therapeutic intervention to co-existing respiratory obstructive syndromes to obtain optimal control of early lung cancer.”  

C2) It is customary to have the limitations paragraph at the end of the Discussion. Currently it is placed as the penultimate paragraph. Please shift it to the end.

A2). We thank the Reviewer for this comment and the paragraph discussing study limitations has been shifted to the end.

C3) While the partition tree analysis identifies smoking and respiratory function as top predictors, I wonder whether the authors considered using a traditional logistic regression model to see if those two factors remained significant predictors after adjusting for all possible confounders.

A3. We thank the Reviewer for pointing out this issue. To determine how much a predictive model is really able to describe reality, the validation methods differ according to whether they are regression, used to estimate a number or classification which associates each element with belonging categories.  While for the regression we need to evaluate the difference between two numbers (observed value and the predicted one), in the classification approach it is sufficient to count the number of times in which the class attributed by the model (our target) is the right one, namely the one observed. Since the cohort evaluated is small sized the regression model cannot be validated.  The text has been implemented accordingly.

This manuscript is a resubmission of an earlier submission. The following is a list of the peer review reports and author responses from that submission.

Round 1

Reviewer 1 Report

1.     Line 69: reduction in life expectancy ïƒ references needed!

2.     All the introduction part must address the newest role of the segmentectomies in the light of the newest CALBG and JCOG studies. 

3.     line 87: atypical segmentectomies ïƒ the authors may provide more info about the type of segments and precisely define the type of typical segmentectomies and atypical ones.

4.     lines 88-89: number of included patients has not to be included in the material and methods section, but in the results; moreover, more accurate and extensive inclusion criteria may be provided (eg respiratory function: which is the range which guided the choice among the surgical resections)

5.     line 97: remove ; 

6.     lines 98-99: imaging examinations do not provide cyto-histologic diagnosis. 

7.     lines 101-105: what do you mean for imaging grading? Disagreement between radiologist about what? Diagnosis? This sentence and its meaning are not clear and the sentence must be rewritten. 

8.     line 104: not were but was

9.     line 111: what does this sentence mean? did you means as instead of has?

10.  lines 153-155: English must be reviewed

11.  line 160: 2.7 ± 1 ïƒ after the 1, the decimal must be provided also here if in the other measures it is included (see 2.6 ± 0.94 and 1.75 ± 0.94)

12.  lines 160-164: This is not a result and has to be discussed in the discussion section

13.  lines 175-190: these results have not to be put in the figure legend, but in the results. Moreover, kaplan meier curves have to be included in the study

14.  lines 194-195: was 28 out of 153 patients (18.3%)after the 18.3, the decimal must be provided also here if in the other measures it is included ( see 16 out of 103 cases in the 194 lobectomy group (15, 53%), 5 out of 26 patients in the typical segmentectomy (19.23%)

15.  line 200: segmentectomy or lobectomy are not a method: please change this word 

16.  lines 200-202: This is not a result and has to be discussed in the discussion section

17.  lines 203-206: results presented in the tables have not to be repeated in the text, also because the text is poorly fluent and it is difficult to follow with all these data in it. 

18.  line 218: what do you mean for free survival from mean disease?  

19.  lines 218-220: all this sentence has to be revised; kaplan meier curves have to be provided because these results are difficult to be read along the text without a figure

20.  line 222: it should be evaluated 

21.  line 244-247: English should be revised!

22.  lines 248-258: these results have to be presented in a more schematic way, and all the comment on the results have to be inserted in the discussion session and not in the result section.

23.  discussion section: all the discussion part has to be better rewritten. All the results have to be better discussed, by comparing them to the newest literature and more references have to be added. In this discussion there only is a repetition of the results. 

24.  lines 276-278, 286-287: these are results;  no need to repeat them in the discussion section 

25.  lines 289-291: English must be revised!

Reviewer 2 Report

I am pleased to have the opportunity to review this paper which contains very interesting and important content for discussion. The authors cite smoking status and respiratory function as predictors of prognosis after early-stage lung cancer surgery, but not tumor histology or stage. While this is an important result if true, it would be difficult to accept it as is for the reasons listed below.

Abstract

1.       The objective clearly states the comparison between lobectomy and segmentectomy, but the conclusion is mainly to discuss prognostic factors. There should be a one-to-one correspondence between objectives and results.

2.       When dealing with the prognosis of stage I lung cancer, a postoperative follow-up period of at least 5 years is required.

Introduction

1.       As the authors describe, there have been clinical trials comparing lobectomy and segmentectomy, and results from trials with a larger number of patients than this paper have already been published, such as the JCOG0802 trial. In light of this, the significance of presenting this paper is not clear.

Materials and Methods

1.       Is TNM the 8th edition?

2.       It is difficult to find a clear definition of the type of case assignment for lobectomy, segmentectomy, and atypical segmentectomy.

Results

1.       There is a description of the number of lymph nodes resected and a significant difference in the number of lymph nodes resected by surgical procedure, but there is no discussion of this. On the other hand, in the case of segmentectomy, if the number of lymph nodes dissected is small, the diagnosis of N1 or N2 may not have been made properly.

2.       Only R0 cases should be included in the analysis.

3.       Insufficient surgical margin is estimated in segmentectomies, which may interfere with the results of clinical trials.

4.       If resection margins are likely to be close together, lobectomy should be selected.

5.       The authors have performed segmentectomy at stage IB, but lobectomy should generally be recommended for tumors larger than 3 cm.

6.       It is important for prognostic analysis to show a survival curve.

Discussion

1.       Patients with squamous cell carcinoma should generally have a poor prognosis because they have a history of heavy smoking and may have multiple other diseases. There is no discussion in this regard.

2. The results of the JCOG study have already been published.  Saji H, Okada M, Tsuboi M, et al. Segmentectomy versus lobectomy in small-sized peripheral non-small-cell lung cancer (JCOG0802/WJOG4607L): a multicentre, open-label, phase 3, randomised, controlled, non-inferiority trial. Lancet. 2022;399(10335):1607–17.